# An Exploration of the Influence of Abrasive Water Jet Pressure on the Friction Signal Characteristics of Fixed Abrasive Lapping Quartz Glass Based on HHT

**DOI:** 10.3390/mi14040891

**Published:** 2023-04-21

**Authors:** Yanling Zheng, Zhao Zhang, Zhankui Wang, Minghua Pang, Lijie Ma, Jianxiu Su

**Affiliations:** 1Henan Institute of Science and Technology, Xinxiang 453003, China; 2Postdoctoral Station, Henan University of Science and Technology, Luoyang 471000, China

**Keywords:** fixed abrasive pad, abrasive water jet, friction, Hilbert marginal spectrum, friction coefficient

## Abstract

Abrasive water jetting is an effective dressing method for a fixed abrasive pad (FAP) and can improve FAP machining efficiency and the impact of abrasive water jet (AWJ) pressure on the dressing effect; moreover, the machining state of FAP after dressing has not been thoroughly studied. Therefore, in this study, the FAP was dressed by using AWJ under four pressures, and the dressed FAP was subjected to lapping experiments and tribological experiments. Through an analysis of the material removal rate, FAP surface topography, friction coefficient, and friction characteristic signal, the influence of AWJ pressure on the friction characteristic signal in FAP processing was studied. The outcomes show that the impact of the dressing on FAP rises and then falls as the AWJ pressure increases. The best dressing effect was observed when the AWJ pressure was 4 MPa. In addition, the maximum value of the marginal spectrum initially rises and then falls as the AWJ pressure increases. When the AWJ pressure was 4 MPa, the peak value of the marginal spectrum of the FAP that was dressed during processing was the largest.

## 1. Introduction

The fixed abrasive machining technique was first proposed in the 1990s. It can quickly lap flat workpieces and has been extensively used in the machining of parts in aerospace and optical communication industries [1,2]. In contrast to the conventional free abrasive lapping technique, the fixed abrasive lapping technique involves fixing the abrasive on pad and wiping the workpiece surface using the exposed abrasive grains on the pad [3,4,5]. As the lapping time continues, the surface of the fixed abrasive pad (FAP) experiences abrasive wear and glazing, making the surface of the FAP flat and smooth, which decreases its processing performance. Therefore, the dressing of the FAP surface is required to improve its machinability.

Abrasive water jet (AWJ) technology is a new processing technology. Its principle is that the abrasion strikes the workpiece surface with the high-speed water jet, thus realizing the machining of the workpiece surface [6,7]. It has strong applicability, high flexibility, low damage, and other characteristics, and has been widely used in surface processing, surface finishing, and other fields in recent years [8,9]. Qu et al. [10] used AWJ to perforate shale with different components, thus defining the damage characteristics of AWJ to shale with different components. Lv et al. [11] used ultrasonic-assisted AWJ to polish aluminum nitride as this method can achieve efficient aluminum nitride processing and high surface quality. Che et al. [12] explored the effect of AWJ process parameters on polishing quality. The experimental results showed that ideal polishing results could be obtained by adjusting the process parameters. Zhu et al. [13] used AWJ for precision surface processing of difficult-to-process materials. The results showed that AWJ has great potential in the processing of difficult-to-machine materials. Zhang et al. [14] used an ultrasonic-assisted micro AWJ processing device to conduct experiments on K9 glass erosion. This technology can realize high efficiency and the high-quality machining of K9 glass using AWJ.

The machinability of an FAP is directly related to the surface state after finishing. Moreover, there is a certain correlation between the surface state of an FAP and the friction signal between its interfaces. Therefore, the machining capability of an FAP can be described by the friction signals between its interfaces. However, friction signals are characterized by instability and nonlinearity, which require appropriate processing and extraction of the characteristic signals. At present, researchers have conducted a lot of research on fault detection and process monitoring. For example, Wada M. et al. [15] investigated the characteristics of the acoustic emission (AE) from frictional wear, observing the correlation between the frictional wear phenomenon and the magnitude of the AE signal, and found that the magnitude of the AE signal was smaller in the wear state and larger in the adhesive wear state. Olalere et al. [16] analyzed the vibration signals in turning machining to classify the working state of the tool and found that the method was more accurate than other models in predicting tool failure by comparison. Pandya et al. [17] used wavelet packets to decompose and extract signals from healthy and faulty bearings and used energy and kurtosis to diagnose bearing faults so that bearing faults could be accurately identified. Liu et al. [18] processed and analyzed the cutting force signal based on the Hilbert–Huang transform (HHT), thus facilitating an accurate monitoring of the tool wear state. Shen et al. [19] monitored the tool state under different working states, which led to the identification of a feature able to characterize the tool worn state that is resistant to interference. Ma et al. [20] monitored the frictional state of bearings under different operating states based on wavelet packets, thus establishing a model for assessing the state of bearings, and the model can accurately identify faults. Chen et al. [21] classified the processing of an FAP based on acoustic emission technique, and the method allows for a better identification of the processing of an FAP for its accurate monitoring. Tyč et al. [22] monitored the process of AWJ cutting hard materials using vibration signals. This method allows for an effective monitoring of the AWJ machining process.

In summary, researchers have conducted a lot of research on AWJ trimming technology and process machining inspection technology, and a lot of results have been achieved. However, no one has studied the effect of AWJ pressure on the FAP dressing effect and machining process by dressing FAP with AWJ. Therefore, this paper uses AWJ with different pressures to dress an FAP surface. Through an analysis of experimental data, the impact of AWJ pressure on the dressing performance of an FAP and the feature signal of friction in a dressed FAP was studied.

## 2. Materials and Methods

### 2.1. Experimental Equipment

The AWJ dressing system was selected to dress the FAP in the experiment. The dressing system was designed and built in the laboratory. Figure 1 shows the working principle of the AWJ. In the selection of abrasives, it was considered that too small an abrasive size would cause an insignificant dressing effect that was not obvious, while too large an abrasive size would easily block the nozzle and cause excessive dressing. Therefore, W3.5 white corundum abrasive was chosen, which has the properties of high hardness and high strength. The name standard is based on the China National Standards: W indicates the particle size, and its unit is μm. Therefore, W3.5 means the abrasive with a particle size of 3.5 μm. A ZDHP-30B lapping and polishing machine (Shenyang Maike material processing equipment Co., Ltd, Shenyang, China) was used for the lapping experiments. Figure 2 shows the processing equipment and schematic diagram. An FAP of type W7 was chosen for the lapping experiments, as shown in Figure 3; the preparation process was taken from the literature [23]. Deionized water was chosen as the lapping slurry. The test workpiece was quartz glass. The test piece is quartz glass, which has high chemical stability, good light transmission and low electrical conductivity, and has a Mohs hardness of 7. Tribological experiments were conducted on an MWF-500 tribological wear tester (Jinan Huaxing Testing Equipment Co., Ltd., Jinan, China). The equipment is shown in Figure 4. Figure 5 shows the experimental process diagram.

### 2.2. Experimental Design

This paper investigates the effect of four AWJ pressures on the frictional signal characteristics of the FAP after dressing. The experimental part consisted of a lapping experiment and a tribological test. Moreover, its tests are carried out three times using a repeatability test to reduce the error. Before the lapping experiment, the AWJ device was used to dress the FAP. Table 1 shows the dressing parameters. Quartz glass lapped the FAP after dressing with AWJ to obtain the arithmetic mean deviation of the roughness profile Ra and material removal rate (MRR) of the workpiece, as well as the surface morphology and Ra of FAP. The tribology test was used to collect the friction signal generated by the FAP sample after dressing with AWJ. The signal processing part mainly processes the friction signal through HHT and obtains the corresponding marginal spectrum peak.

### 2.3. Feature Selection and Extraction of Friction Signal

Recent studies have shown that the HHT (Hilbert–Huang transform) method, which is commonly used in nonstationary signal processing, has high adaptability and can better represent the variation of the original signal during the analysis and processing [24,25]. Therefore, the Hilbert marginal transform that extracts the friction signal while lapping the process of the FAP can accurately represent the changes of the signal, which is combined with the surface situation of the FAP. The HHT method includes two steps: EMD (Empirical Mode Decomposition) and Hilbert transform [26]. First, EMD is used to decompose the signal to obtain a limited number of the Intrinsic Mode Function (IMF). Then, the Hilbert transform is performed on each feature mode IMF to obtain the Hilbert spectrum. Figure 6 shows the feature extraction process.

### 2.4. Measurement and Characterization

#### 2.4.1. Workpiece MRR

After each group of lapping tests, the workpiece samples were placed in an ultrasonic cleaner for 5 min to remove impurities from the surface of the workpiece samples. The ultrasonic cleaned and dried samples were placed in a precision balance (Satorious CP225D) and the sample mass was measured and the MRR was calculated using Equation (1):(1)$$\mathrm{MRR}=\frac{\Delta m}{\rho \times s\times t}\times 10^{7}$$

#### 2.4.2. Workpiece Ra and 3D Morphology

The surface topography and Ra of the workpiece sample after ultrasonic cleaning are measured using a Contour GT-X3/X8 white light interferometer (Bruker, Nano, Inc., Berlin, Germany) which has a measurement accuracy of 0.001 nm. To reduce measurement errors, three points are selected uniformly on the surface of the sample for each test.

#### 2.4.3. Surface State and 3D Morphology of FAP

The processed and treated FAP samples were fixed on the stage and a blower was used to remove impurities and dust from the FAP surface. Then, the surface topography was measured using a contour white light interferometer (Contour GT-X3/X8) to obtain the Ra and 3D roughness parameters of FAP: the maximum height of the summit Sp, skewness of the height distribution Ssk, kurtosis of the height distribution Sku, and root mean square deviation of the surface Sq. Based on the existing reports, the Sp represents the height of abrasive grains on the surface of FAP, the Sq can represent the state of surface texture, the Ssk represents the surface symmetry, and the Sku represents the surface height distribution [27]. Therefore, the above roughness parameters are chosen to accurately represent the change in the surface condition of the FAP.

## 3. Results and Discussion

### 3.1. Experimental Results

Figure 7 shows the MRR and Ra of the FAP after dressing using AWJ with four pressures. Figure 7 shows that the MRR of the FAP dressed using a 4 MPa pressure AWJ is greater than other AWJ-dressed pressures, while the surface quality obtained by lapping the workpiece with the FAP dressed with a 5 MPa pressure AWJ is the smallest.

### 3.2. Effect of AWJ Pressure on Surface Morphology after FAP Dressing

Figure 8 shows the surface parameters of the FAP after dressing with AWJ under four pressures. Moreover, Figure 9 shows the morphology of the FAP after dressing with four AWJ pressures. It can be seen from Figure 8 that the Sp, Sq, and Ssk of the FAP after lapping are the smallest, while Sku is the largest. With an increase in AWJ pressure, Sp, Sq, and Ssk increase firstly and then decrease, whereas Sku decreases firstly and then increases. This is due to the small number and low height of the abrasive grains exposed on the FAP surface. Most of the substrate is exposed to the surface, which is relatively smooth, and the Ra is the smallest. The AWJ pressure is 2 MPa, and it does not provide enough energy for the abrasive to remove the protruding matrix and worn abrasive grains from the FAP surface while dressing. The exposed height of the abrasive grains on the surface is small, and the phenomenon of high local concentration is not well improved. Therefore, Sp, Sq, and Ssk are small, while Sku is large. With an increase in the pressure of the AWJ, the abrasive sprayed by the nozzle can obtain greater kinetic energy. The abrasive can effectively remove the surface matrix and worn abrasive grains. It also improves the local height concentration and exposes more fresh abrasive grains on the surface of the FAP. When the AWJ pressure is 4 MPa, its Sp, Sq, and Ssk are the largest, whereas Sku is the smallest, which shows that, when the pressure is 4 MPa, the AWJ has the best dressing performance for the FAP, in which the exposed height of the surface abrasive grains is the largest and its Ra is the largest. When the AWJ pressure is 5 MPa, due to the rapid flow rate in the mixing chamber, some abrasive grains will be broken, which will affect the dressing effect of the AWJ on the abrasive pad; therefore, Sp, Sq, and Ssk decrease and Sku increases.

### 3.3. Effect of AWJ with Four Pressures on Machining Performance of FAP and Surface Morphology of Workpiece

Figure 10 shows the workpiece morphology of FAP lapping with AWJ dressing under four pressures. Figure 11 shows the Ra of FAP and workpiece after dressing with AWJ under four pressures. Figure 11 shows that the Ra of the FAP rises and then falls with an increase in AWJ pressure, while the Ra of the workpiece firstly decreases and then increases with a rise in AWJ pressure. This is because, when the AWJ pressure is 2 MPa, the AWJ cannot effectively improve the surface state of the FAP. The local higher matrix or abrasive grains contact and scratch the workpiece. The abrasive grains cut into the surface deeply, and the scratch caused by scratching is deep. The Ra of the workpiece is the largest, and the Ra of the FAP is small. With an increase in AWJ pressure, the kinetic energy obtained by abrasive increases, which can effectively improve the surface state of the FAP. When the AWJ pressure is 4 MPa, the exposed height of the abrasive grains increases, and the greater the Ra of FAP, the better the machining performance. During the lapping process, more abrasive grains make contact with the workpiece. The abrasive grains are subjected to small and uniform forces, resulting in small scratches caused by scraping and the good surface quality of the workpiece (Figure 9). When the AWJ pressure is 5 MPa, the dressing performance of the abrasive will be affected in the mixing room. The effect of the phenomenon of local height concentration on the surface of the FAP leads to a large and uneven force on abrasive grains in the machining process, which contributes to the scratches caused by abrasive grains scratching the workpiece becoming deeper, and the workpiece’s surface quality worsens as a result. Therefore, when the AWJ pressure is 4 MPa, the FAP machining performance and workpiece surface quality are the best after dressing.

### 3.4. The Influence of AWJ Pressure on Friction Coefficient of FAP

The friction coefficient has a certain relationship with the wear degree and micro cutting performance of FAP surface abrasive grains [28]. Therefore, the coefficient of the friction collected can indirectly represent the surface state and machinability of the FAP after four AWJ dressing pressures. Figure 12 shows that the coefficient of the friction of the FAP dressed using AWJ at four pressures. From Figure 12b, the coefficient of friction first rises and then falls with an increase in AWJ pressure. When the AWJ pressure is 4 MPa, the coefficient of the friction of the FAP during machining is the largest; however, when the AWJ pressure is 2 MPa, the coefficient of the friction of the FAP during machining is the smallest. This is because the AWJ with 4 MPa pressure has the best dressing effect on the FAP. The matrix and worn abrasive grains on the surface of the FAP can be effectively removed during the dressing process, which greatly enhances the machinability of the FAP. More abrasive grains on the surface of the FAP contact and scratch the workpiece to maximize the friction coefficient. The AWJ under a 2 MPa pressure has the worst dressing performance on the FAP, as this leads to an ineffective removal of the matrix and abrasive grains from the FAP surface during dressing, making its plus performance poor. The FAP surface has better matrix contact and scrapes the workpiece, thus minimizing the coefficient of friction during machining.

### 3.5. Effect of AWJ with Different Pressure on Friction Characteristic Signal of FAP

The friction signal is processed using MATLAB software. A marginal spectrum of the FAP after dressing with four AWJ pressure was obtained. Figure 13 shows that the maximum value of the marginal spectrum is mainly around 4.7 Hz and that it is related to the cutting ability of the FAP [29]. Therefore, the processability of the FAP after dressing with AWJ under four pressures can be reflected by the maximum value of the marginal spectrum.

Figure 14 shows the relationship between the maximum value of the edge spectrum and the AWJ pressure, and its value first rises and then falls as the AWJ pressure increases. This is because the surface state of the FAP is not effectively improved at AWJ pressures of 2 MPa; moreover, because the worn particles and protruding matrix on the FAP surface are not effectively removed, the matrix mainly scrapes the workpiece to produce its frictional characteristic signal. Therefore, the edge spectrum peak is minimal. The surface state of the FAP was effectively improved when the AWJ pressure was 4 MPa. Abrasive grains and the excess matrix worn on the FAP surface can be effectively removed. The abrasive grains mainly contact and scrape the workpiece to achieve material removal. Therefore, the edge spectrum peak energy is the maximum. AWJ pressure is 5 MPa, and the dressing performance of the abrasive on the FAP is affected. The abrasive particles and excess matrix on the FAP surface cannot be effectively removed, and there is more of the matrix on the FAP surface. The more of the matrix that touches the workpiece during machining, the more the peak marginal spectral value decreases.

### 3.6. Discussion

In AWJ dressing, the pressure of the AWJ is an essential factor in the machinability of the FAP after dressing. Figure 14 shows a schematic diagram of the AWJ dressing. Figure 15a shows that, after the abrasive and water are mixed in the mixing room, the mixture is sprayed through the nozzle. The interaction between abrasive and water removes the matrix and abrasive from the FAP surface and enhances its machinability. When the pressure of AWJ is 2 MPa, the abrasive and water are not fully mixed in the mixing room, and the kinetic energy of the abrasive and water ejected from the nozzle is small. It is difficult to remove the matrix and worn abrasive grains from the FAP surface, and the dressing performance is poor. As the pressure of AWJ increases, the abrasive and water can be fully mixed. When the AWJ pressure is 4 MPa, the kinetic energy of the abrasive and water ejected from the nozzle is large. As shown in Figure 15b, it can effectively remove the matrix and abrasive grains of wear from the FAP surface, improve the local high concentration of the surface, and possess a good dressing effect. As shown in Figure 15c, when the pressure of AWJ is 5 MPa, it will accelerate the flow of abrasive in the mixing room, so that more abrasive will be broken in the mixing room and thus more broken abrasive will be ejected from the nozzle, which will eventually lead to a decrease in the removal of the matrix and abrasive grains on the surface of the FAP.

Regarding the FAP surface, by dressing with AWJ, the excess matrix and worn abrasive grains on the FAP surface are removed, and more fresh abrasive grains are exposed. When the AWJ pressure is 2 MPa, the AWJ is not effective in dressing the FAP surface, the abrasive grains on the FAP surface cannot be effectively removed, and its machining capability cannot be effectively enhanced. There is still a significant amount of worn abrasive grains in contact with the workpiece and scraping it. At this time, the marginal spectral peak value generated by abrasive grains scratching the workpiece is the smallest (Figure 16b). When the AWJ pressure is 4 MPa, the worn abrasive grains and the redundant matrix on the FAP surface can be effectively removed, and the FAP surface after AWJ dressing has more fresh and sharp abrasive grains. Fresh abrasive grains can be easily cut to the workpiece. The large removal thickness of the workpiece makes its MRR large (Figure 8a). When fresh abrasive scratch and cut the workpiece, a significant amount of debris is generated. Under the influence of the flow of the lapping slurry, the debris will form eddy currents between the FAP and the workpiece, and the abrasive grains will scratch the workpiece and matrix off the FAP under the drive of the eddy currents [30]. The shedding of worn particles and the appearance of fresh abrasive grains are promoted to achieve the self-repair of the FAP [31]. Therefore, the maximum value of the marginal spectrum is the largest (Figure 16c). When the AWJ pressure is 5 MPa, the dressing performance of the AWJ decreases, the removal rate of worn abrasive grains and the exposure rate of fresh abrasive grains during lapping slows down, and the amount of fresh abrasive grains scraping the workpiece decreases. As a result, the maximum value of the marginal spectrum is reduced.

## 4. Conclusions

In the present work, dressing tests of the FAP were conducted using a self-designed AWJ system, and separate lapping tests and tribological tests of the dressed FAP were conducted. The influence of jet pressure on the FAP morphology, workpiece morphology, MRR, and frictional characteristic signals was discussed, and the mechanism of the influence was analyzed. The results are as follows:

An AWJ-dressed FAP can effectively improve its machining performance, and the AWJ pressure greatly impacts the dressing effect. The Ra of the FAP firstly rises and then falls with the AWJ pressure. When the AWJ pressure is 4 MPa, the dressing effect of the AWJ is the best, the machining performance of the FAP is the best, and the maximum MRR can reach 632.76 nm/min.

The AWJ pressure has an impact on the friction coefficient of the dressed FAP. The friction coefficient increases firstly and then decreases with an increase in AWJ pressure. When the FAP was dressed with 4 MPa AWJ, the micro-cutting capability of the FAP surface abrasive grains was the strongest and the friction coefficient was the largest, up to 0.184.

The maximum value of the marginal spectrum can be used to detect the micro cutting capability of FAP surface abrasive grains, and the AWJ pressure has a certain influence on the maximum value of the marginal spectrum generated during the processing of the dressed FAP. The value first rises and then falls as the AWJ pressure increases. When the AWJ pressure is 4 MPa, the value is at its largest.

## Figures and Tables

**Figure 1 micromachines-14-00891-f001:**
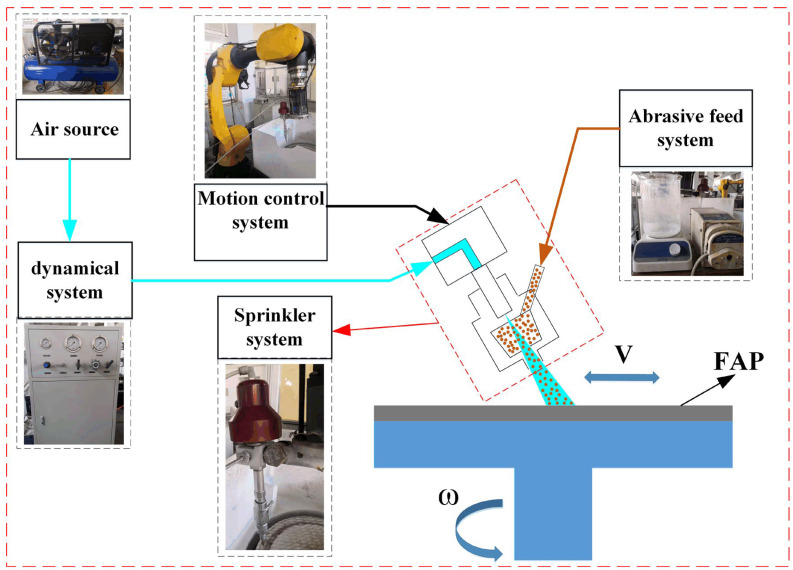
Working principle of AWJ.

**Figure 2 micromachines-14-00891-f002:**
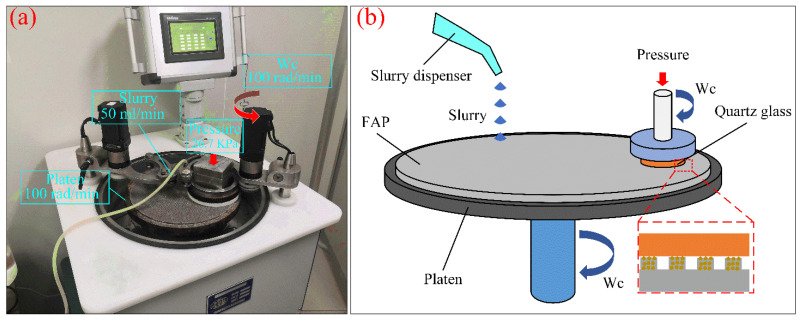
ZDHP-30B lapping and polishing machine and lapping schematic diagram: (**a**) test equipment; (**b**) working principle diagram.

**Figure 3 micromachines-14-00891-f003:**
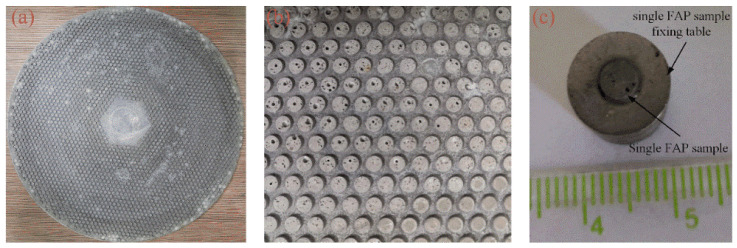
Polishing pad measurement sample: (**a**) FAP; (**b**) FAP local magnification; (**c**) FAP sample.

**Figure 4 micromachines-14-00891-f004:**
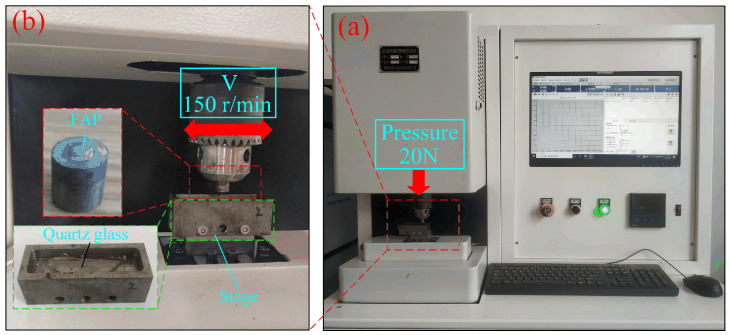
MWF-500 equipment and local amplification: (**a**) equipment working position; (**b**) test equipment.

**Figure 5 micromachines-14-00891-f005:**
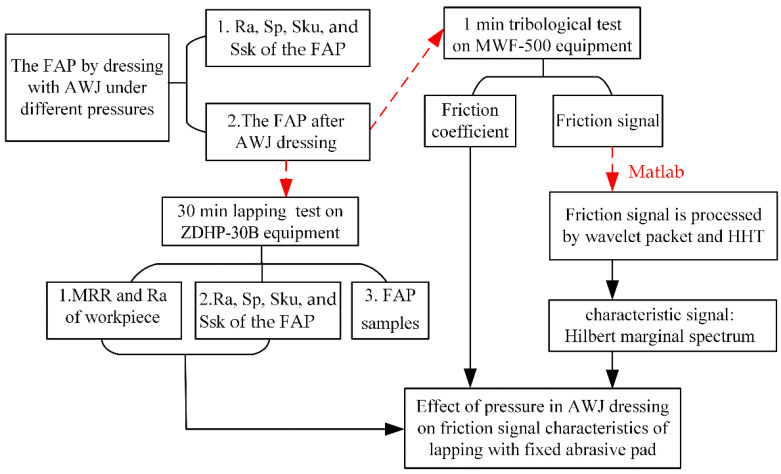
Experimental process diagram.

**Figure 6 micromachines-14-00891-f006:**
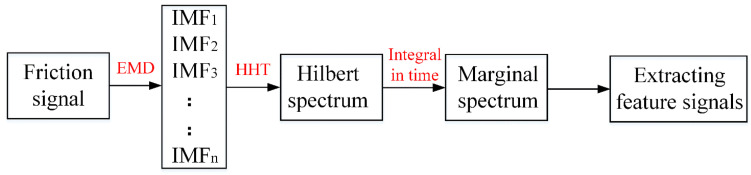
The feature extraction process.

**Figure 7 micromachines-14-00891-f007:**
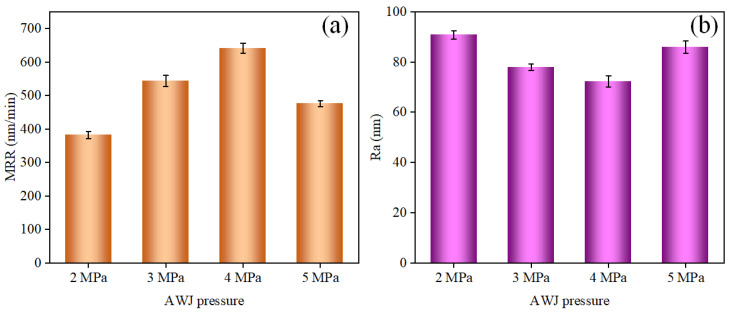
MRR and Ra of the FAP after dressing using AWJ with four pressures: (**a**) MRR, (**b**) Ra.

**Figure 8 micromachines-14-00891-f008:**
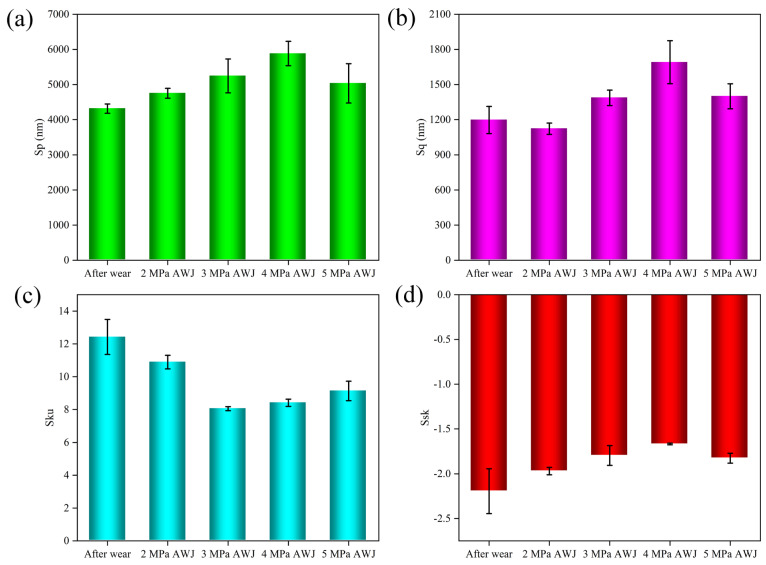
Surface parameters of FAP after dressing with AWJ under four pressures: (**a**) Sp; (**b**) Sq; (**c**) Sku; (**d**) Ssk.

**Figure 9 micromachines-14-00891-f009:**
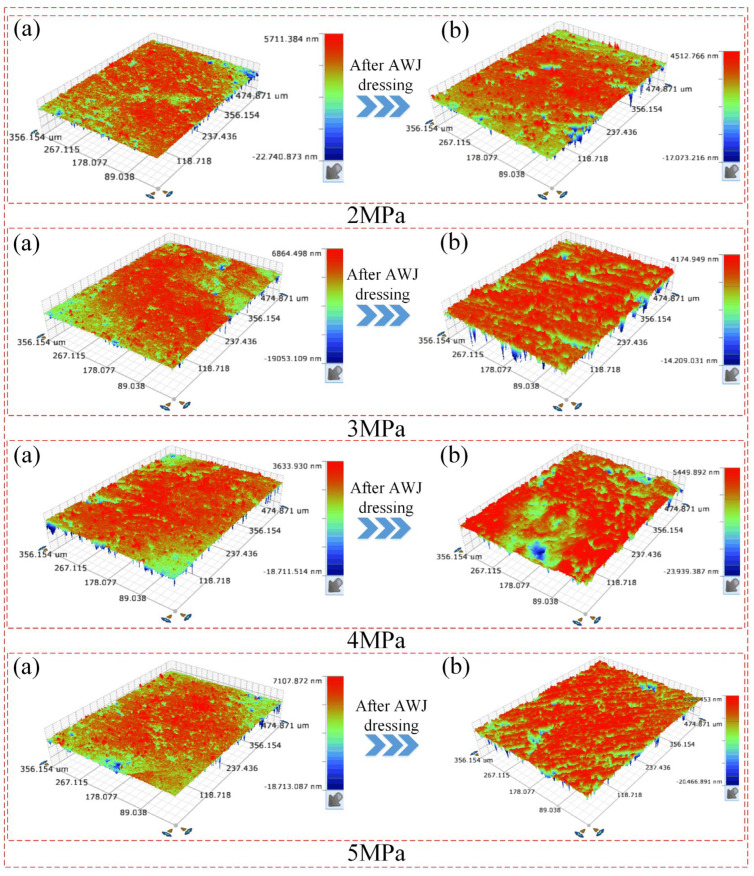
Morphology of FAP before and after dressing with AWJ under four pressures: (**a**) after lapping; (**b**) after AWJ dressing.

**Figure 10 micromachines-14-00891-f010:**
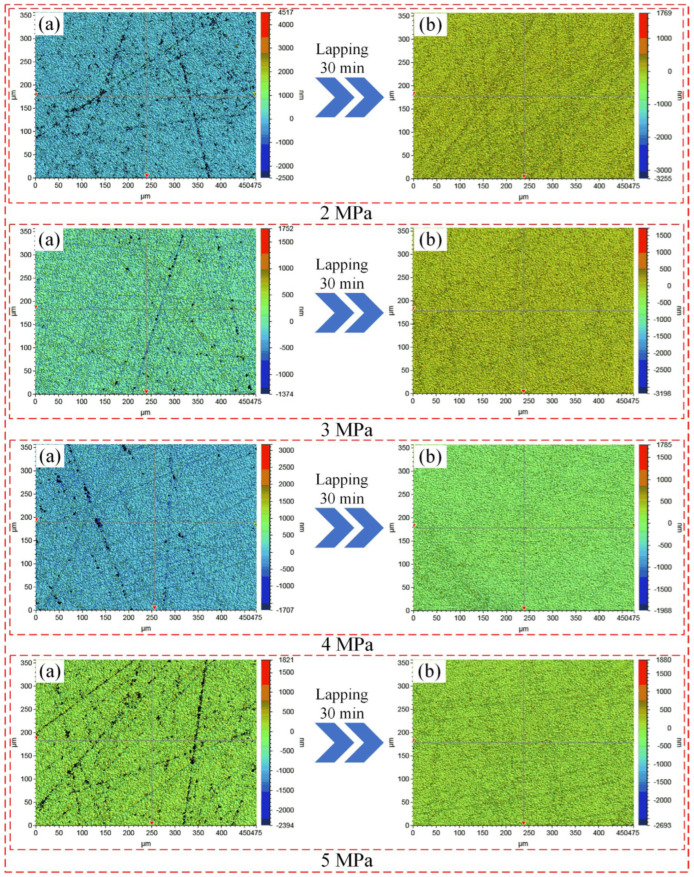
Workpiece morphology of FAP lapping with AWJ dressing under four pressures: (**a**) before lapping; (**b**) after lapping.

**Figure 11 micromachines-14-00891-f011:**
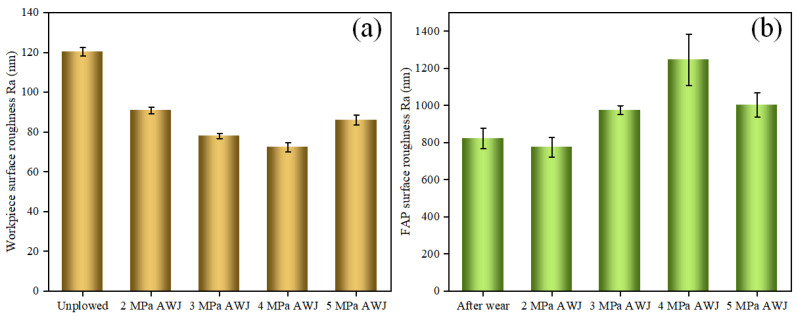
Ra of FAP and workpiece after dressing with AWJ under four pressures: (**a**) Ra of workpiece; (**b**) Ra of FAP.

**Figure 12 micromachines-14-00891-f012:**
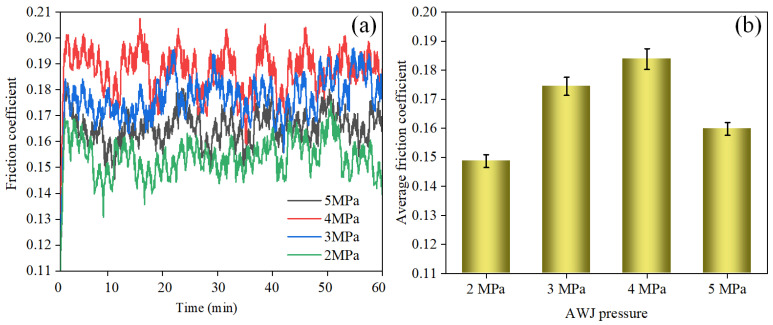
Friction coefficient for AWJ-dressed FAP at four pressures: (**a**) friction coefficient; (**b**) average friction coefficient.

**Figure 13 micromachines-14-00891-f013:**
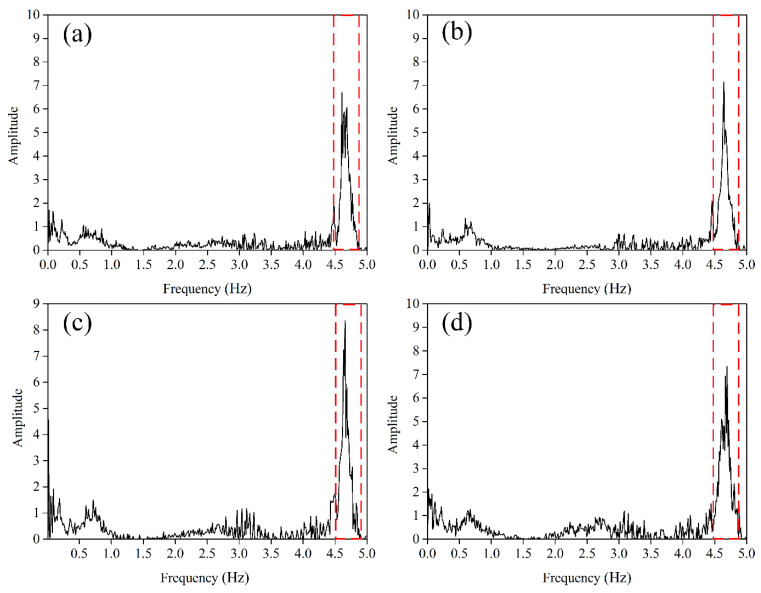
Marginal spectrum maximum value of FAP after dressing with AWJ under four pressures: (**a**) 2 MPa; (**b**) 3 MPa; (**c**) 4 MPa; (**d**) 5 MPa.

**Figure 14 micromachines-14-00891-f014:**
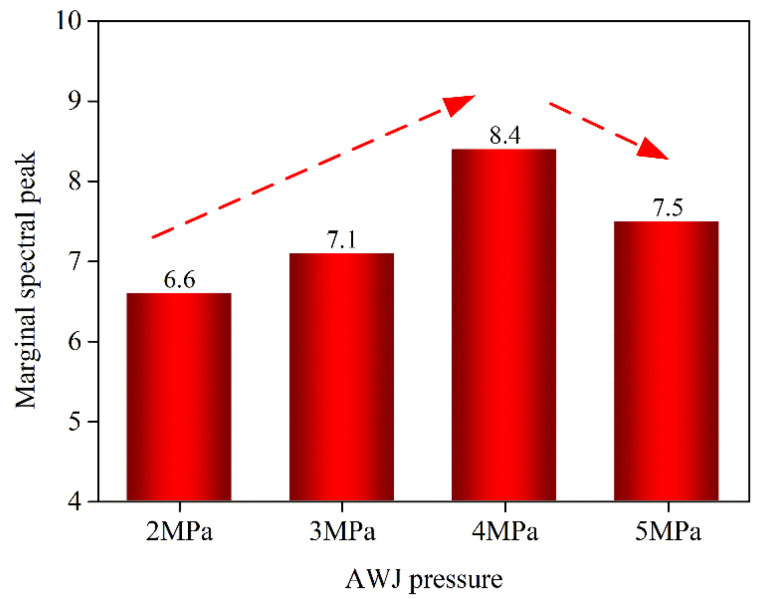
The relationship between the maximum value of the marginal spectrum and the AWJ pressure.

**Figure 15 micromachines-14-00891-f015:**
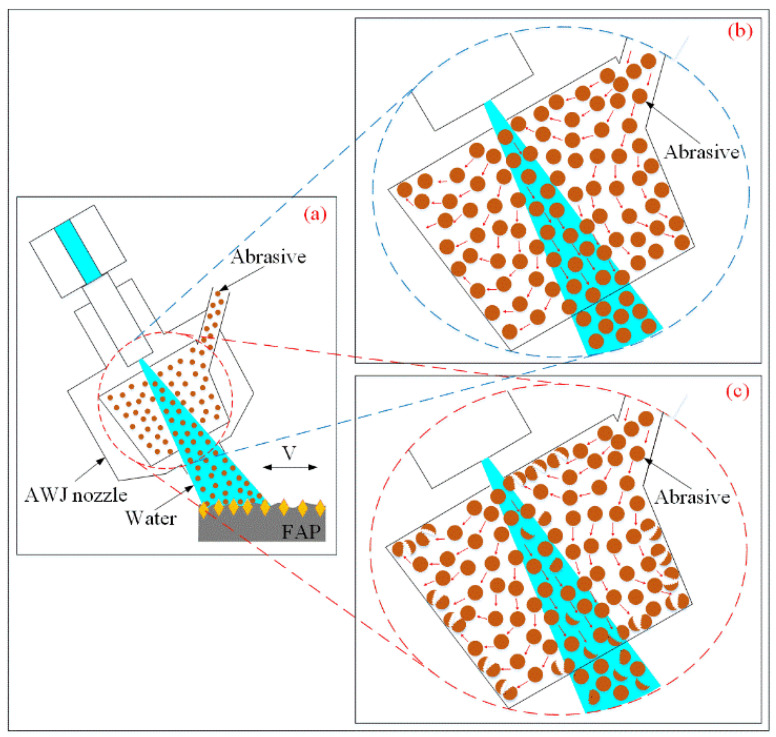
Schematic diagram of AWJ dressing the FAP: (**a**) schematic diagram of AWJ dressing; (**b**) 4 MPa AWJ dressing diagram; (**c**) 5 MPa AWJ dressing diagram.

**Figure 16 micromachines-14-00891-f016:**
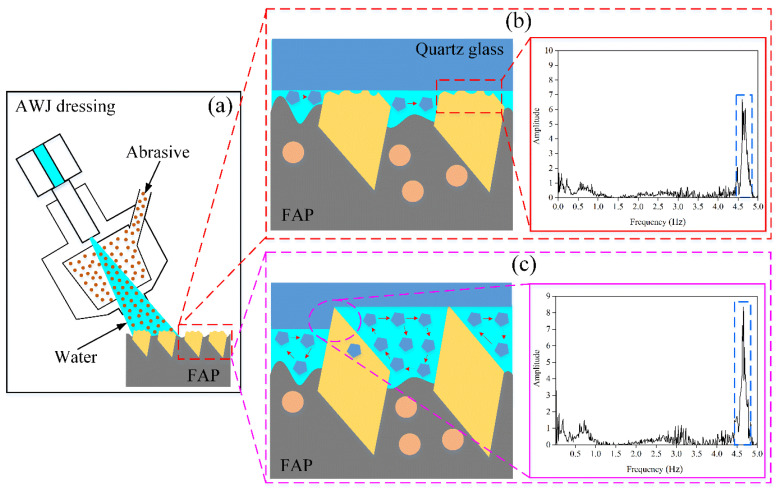
Lapping mechanism of FAP: (**a**) schematic diagram of AWJ dressing; (**b**) 2 MPa; (**c**) 4 MPa.

**Table 1 micromachines-14-00891-t001:** Process parameters of AWJ dressing test.

Abrasive Size	Abrasive Flow	Rotational Speed	Traverse Speed	Dress Time	AWJ Pressure
W3.5 (3.5 μm)	100 cm^3^/min	110 r/min	2.5 mm/s	5 min	2 MPa, 3 MPa, 4 MPa, 5 MPa

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
