# Peer review of "An Exploration of the Influence of Abrasive Water Jet Pressure on the Friction Signal Characteristics of Fixed Abrasive Lapping Quartz Glass Based on HHT"

_micromachines, 2023, doi:10.3390/mi14040891_

Round 1

Reviewer 1 Report

The authors published using of Abrasive water jet for effective dressing method of fixed abrasive pad (FAP) in improve FAP machining efficiency aspect. The paper is interesting, although restoring the cutting ability of abrasive tools by high-pressure water jet is not very novel. But the big weakness of this paper is roughness measurements without specifying the measurement conditions and the used filters. In this form, the article is useless.

  Noticed errors and comments

1.       Chapter 2 is called Materials and methods. And where is the information about these materials? The authors did not characterize the materials used in the study at all, i.e., quartz glass and abrasive material and did not give any of their properties.

2.       Table 1. What is W3.5 abrasive size? From what standard did the authors adopt this designation? What is the unit of size of this abrasive?

3.       Table 1. The liter is not a legal unit in the SI. The legal unit of volume in SI is m3 and, of course, its aliquots, like cm3. You should change ml to cm3 or mm3.

4.       Table 2 and Figure 7 present identical data and therefore one of these items is redundant.

5.       Figure 7, Figure 8, Figure 11, Figure 12b. No confidence intervals marked.

6.       I propose to rename subsection 3.2 from: “Effect of AWJ with four pressures on surface morphology after FAP dressing” to: “Effect of AWJ pressure on surface morphology after FAP dressing”

7.       The authors did not explain what Ra means. Not all readers are clairvoyant to guess if it's a roughness factor.

8.       My doubt is the use of this roughness profile parameter for samples before machining versus surface roughness parameters (Sa, Sp etc.) after machining. Why didn't the authors use the same parameter in both cases?

9.       Figure 9 is inferior quality and needs improvement.

Small errors

1.       Table 2. Is: 2MPa; should be: 2 MPa. Refers to analogous errors throughout the work.

2.       Pages 11/12 and 15/16. Bad pagination.

Author Response

Dear reviewer:

I am very grateful to your comments for the manuscript. According with your advice, we amended the relevant part in manuscript and marked in red color in the revised manuscript. Some of your questions were answered below.

Author Response

(The authors gave the same response as above.)

Round 2

Reviewer 1 Report

The authors clarified my doubts and corrected the noted errors. Thus, the obstacles to publishing this paper disappeared.